# Covalent Adaptable Network of Semicrystalline Polyolefin Blend with Triple-Shape Memory Effect

**DOI:** 10.3390/polym16192714

**Published:** 2024-09-25

**Authors:** Hann Lee, Yujin Jang, Young-Wook Chang, Changgyu Lim

**Affiliations:** 1Department of Chemical Engineering, Hanyang University, Ansan 15588, Republic of Koreaserapina1027@hanyang.ac.kr (Y.J.);; 2BK21 FOUR ERICA-ACE Center, Hanyang University, Ansan 15588, Republic of Korea

**Keywords:** covalent adaptive network (CAN), maleated PP, maleated POE, blend, reprocessability, triple-shape-memory effects

## Abstract

A covalent adaptable network (CAN) of semicrystalline polyolefin blends with triple-shape memory effects was fabricated by the reactive melt blending of maleated polypropylene (mPP) and maleated polyolefin elastomer (mPOE) (50 wt/50 wt) in the presence of a small amount of a tetrafunctional thiol (PETMP) and 1,5,7-triazabicyclo [4,4,0]dec-5-ene (TBD). The polymer blend formed a chemically crosslinked network via the reaction between the thiol group of PETMP and maleic anhydride of both polymers in the blend, which was confirmed by FTIR, the variation of torque during the melt mixing process, a solubility test, and DMA. DSC analysis revealed that the crosslinked polyolefin blends show two distinct crystalline melting transitions corresponding to each component polymer. Improved tensile strength as well as elongation at break were observed in the crosslinked blend as compared to the simple blend, and the mechanical properties were maintained after repeated melt processing. These results suggest that thermoplastic polyolefin blends can be transformed into a high-performance and value-added material with good recyclability and reprocessability.

## 1. Introduction

A thermally driven shape-memory polymer (SMP) is an important smart material which can have a stably fixed temporary shape and return to its permanent shape from a temporarily fixed shape upon heating above a switching temperature [1,2,3,4,5]. Due to facile tunability in mechanical properties and switching temperatures by a proper molecular design as well as low cost and easy processability, the SMPs can have diverse applications such as biomedical devices [6], 4D printing [7], smart windows [8], and flexible electronic devices [9].

The shape memory effects can be actualized for various crosslinked polymer networks containing reversible phases with a suitable phase-switching temperature (*T*_sw_) which is either the glass transition (*T*_g_) for amorphous polymer networks or the crystalline melting transition (*T*_m_) for semicrystalline polymer networks. The crosslinked polymers are deformed to a desired shape upon heating above its *T*_g_ or *T*_m_, and the shape can be fixed in its deformed state through vitrification or crystallization processes upon cooling. The temporarily fixed shape obtained by this temperature programming can be recovered to its original shape through entropy increase in polymer chains when it is heated above its *T*_sw_.

Conventional SMPs show a dual-shape memory effect, having only one programmed temporary shape which can return to its original shape in one step. Multi-shape-memory polymers that can have more than one temporary shape and return to their original shape step-by-step have received attention [10,11,12,13]. Such multi-shape memory effects can be observed in a multiphase polymer network having distinctly different phase transition temperatures. More specifically, Lendlein et al. showed that triple-shape memory effects can be manifested in a polymer network with two well-separated phase transition temperatures [10,11]. Based on this concept, various types of phase-separated polymer networks demonstrating the triple-shape-memory effects have been designed, which include graft or block copolymers [14,15,16,17], interpenetrating polymer networks [18,19,20], and chemically crosslinked immiscible polymer blends [21,22,23,24,25,26].

In recent years, covalent adaptable networks (CANs) based on associative dynamic covalent chemistries such as transesterification, vinylogous urethane exchange, silyl ether exchange, disulfide metathesis, or oxaborolane metathesis have been paid great attention as new types of thermosetting polymers [27,28,29,30,31]. Unlike conventional thermosets, CANs can undergo topological rearrangement via thermally activated exchange reactions of the dynamic covalent bonds throughout the network without a loss in the crosslink density, which allows these networks to have thermal reprocessability and recyclability while maintaining performance properties of conventional thermosets.

Numerous studies on CAN-based SMPs have been conducted [32,33,34,35,36,37,38,39]. Contrary to the traditional thermoset SMPs, the CAN-based SMPs have several advantages due to the topological rearrangement of their dynamic covalent networks, which include the ability of shape reconfiguration, enabling complex shape changes as well as reprocessability and remoldability. Additionally, shape fixing in the CAN-based SMPs can be achieved by vitrification or crystallization like traditional SMPs [35,36]. However, the fabrications of CAN-based SMPs reported so far are not practically feasible because an elaborate synthesis of new polymers and crosslinkers is needed for the introduction of dynamic covalent bonds into a polymer network.

In this study, we proposed a simple way to fabricate the CAN-based SMPs, demonstrating a triple-shape-memory effect and shape reconfigurability by the simple melt blending of commercially available semicrystalline polyolefins and a crosslinking agent, to determine whether this approach can be practically useful for the upcycling of disposed commodity polyolefins generated abundantly in daily life.

## 2. Materials and Methods

### 2.1. Materials and Sample Preparation

Maleated polypropylene (mPP, maleic anhydride content 1 wt%, Adpoly BP402) was procured from Lotte Chem (Daejeon, Korea) and maleated polyolefin elastomer (mPOE, maleic anhydride content 1 wt%, Fusabond N416) was obtained from DuPont (Wilmington, DE, USA). Pentaerythritol tetrakis(3-mercaptopropionate) (PETMP, >90.0%), 1,5,7-triazabicyclo [4.4.0]dec-5-ene (TBD, 95%), and xylene were purchased from Sigma-Aldrich (St. Louis, MO, USA).

The mPP/mPOE (50/50 wt/wt) blend with different contents of PETMP (0.5, 1, and 2 parts per hundred (phr) of polymers) in the presence of 1 phr of TBD was prepared by melt blending in a Haake mixer (Haake Polylab Rheomix 600, Karlsruhe, Germany) for 15 min at 180 °C with a rotor speed of 60 rpm. The obtained mixtures were compression-molded into films with a uniform thickness of about 1 mm by using a hot-press at 180 °C for 10 min.

### 2.2. Characterization

FTIR analysis was carried out using a Bucker ALPHA spectrometer (Nicolet IS10, Thermo Scientific, Waltham, MA, USA), equipped with an attenuated total reflectance (ATR) accessory, under a nitrogen atmosphere to examine the occurrence of the reaction between maleic anhydride in the mPP/mPOE blend and thiol in PETMP.

Dynamic mechanical tests were carried out by using a dynamic mechanical analyzer (TA Instrument, model DMA-Q800, New Castle, DE, USA). Samples were subjected to cyclic tensile strain with an amplitude of 0.2% at a frequency of 1 Hz. The temperature was increased at a heating rate of 10 °C/min over the range from −100 to 200 °C.

Phase transition temperatures and enthalpy changes of the samples were examined by using a differential scanning calorimeter (TA instruments, DSC Q20 equipped with RSC90 as a refrigerated cooling system, New Castle, DE, USA). The sample was heated from room temperature to 200 °C at a rate of 10 °C/min under a nitrogen atmosphere and was kept for 5 min at this temperature. Next, the sample was cooled down to −50 °C at a cooling rate of 10 °C/min, followed by reheating to 200 °C at a rate of 10 °C/min to monitor the crystallization and crystalline melting transition of the sample, respectively.

Tensile tests for the samples were conducted using a universal testing machine (UTM, AGS-500NX, Shimadzu, Japan) with a crosshead speed of 50 mm/min at room temperature. The tests were repeated until reproducible results were obtained.

For the evaluation of the recyclability of crosslinked samples, the original samples which were obtained by melt mixing and compression molding as mentioned above, were cut into small pieces and compression-molded at 180 °C under a pressure of 15 MPa for 15 min to obtain a sheet with a thickness of about 2 mm, and then the tensile properties of the recycled sample were measured. This process was repeated 3 times under the same conditions.

The triple-shape-memory effects of the samples were measured using DMA, according to the protocol used in our previous paper [25]. Briefly, a rectangular sample (permanent shape, shape A) was deformed at 140 °C (which is above *T*_m_ of the PP and POE phases in the sample) by applying a stress of 0.64 MPa, and then cooled to 80 °C (which is below *T*_c_ of the PP phase and above *T*_m_ of the POE phase in the sample) under a release of the imposed stress to obtain a first temporary shape (shape B). The load (1.0 MPa) was imposed on the sample (shape B), equilibrating at 80 °C to elongate it, followed by cooling to 0 °C (below *T*_c_ of the POE phase in the sample) to obtain a second temporary shape (shape C). The obtained temporarily fixed specimen (shape C) was heated to 80 °C (above *T*_m_ of the POE phase), followed by further heating to 140 °C (above *T*_m_ of the PP phase) with a heating rate of 2 °C/min to induce the recovery from shape C to shape B, and from shape B to shape A, respectively. This thermomechanical cycle was performed consecutively two more times on the same specimen. Shape fixity obtained for shapes B and C, *R*_f_(B) and *R*_f_(C), was determined according to Equations (1) and (2), respectively. Additionally, shape recoveries for the shape changes occurring from C to B, *R*_r_(C→B), and from B to A, *R*_r_(B→A), were determined according to Equations (3) and (4), respectively.
*R_f_*(B) % = (ε_B_/ε_B, load_) × 100(1)
*R_f_*(C) % = (ε_C_/ε_C, load_) × 100(2)
*R*_r_(C→B) (%) = [(ε_C_ − ε_B,rec_)/(ε_C_ − ε_B_)] × 100(3)
*R*_r_(B→A) (%) = [(ε_B_ − ε_A,rec_)/(ε_B_ − ε_A_)] × 100(4)
where ε_B(or C), load_ and ε_A(or B or C)_ stand for the strain under the applied tensile load and after release of the load, respectively. ε_B,rec_ and ε_A,rec_ stand for the obtained strain after recovery to shape B and shape A, respectively.

## 3. Results and Discussion

### 3.1. Formation of Crosslinked Polymer Blend Network

Figure 1a shows the FTIR spectrum of the neat mPOE/mPP blend and the blend crosslinked with PETMP. The neat blend shows absorption bands at 1787 cm^–1^ and 1863 cm^–1^, which are attributed to the stretching vibration of maleic anhydride of mPP and mPOE. In the FTIR spectra for the blend with PETMP, a new absorption peak at 1647 cm^–1^ appeared, attributed to the C=O stretching of thioester bonds while absorption peaks corresponding to maleic anhydride disappeared. This indicates that a reaction occurred between the thiol group of PETMP and the maleic anhydride groups of both polymers, generating thioester bonds which can lead to the crosslinking of polymer chains [30].

Figure 1b shows the torque evolution as a function of time during melt mixing, which was recorded after the polymers were melted completely. For the neat blend, there were no changes in torque with further mixing while the torque increased abruptly when PETMP and a catalyst were added into the polymer melt. Such an increase in the torque implies that the crosslinking of the polymer chains occurred via the thiol–anhydride reaction during the melt mixing process, which resulted in the restriction of polymer chain movement.

The formation of a chemically crosslinked network in the blend with PETMP was also evidenced by a solubility test using hot xylene as a solvent, as shown in Figure 1c. The simple thermoplastic mPP/mPOE blend was dissolved completely in the solvent, while the blend with PETMP was insoluble but swollen in the same condition, confirming that the mPP/mPOE blend can form a crosslinked network by melt mixing the polymers with PETMP.

The chemical structure of the crosslinked network prepared by the crosslinking of the mPP/mPOE blend with PETMP in the presence of a TBD catalyst is shown in Figure 2, in which thiol–thioester exchange reactions can occur in this network upon heating.

### 3.2. DSC Thermograms

Figure 3 shows a DSC thermogram of the samples, and melting temperature (*T*_m_), enthalpy accompanied by the crystalline melting transition (Δ*H*_m_), and crystallization temperature obtained from the thermograms are listed in Table 1. As illustrated in Figure 3a, all blend samples have endothermic peaks at two distinctly different positions, one at around 50.4~52.8 °C, corresponding to the melting of the POE phase and the other one at around 120~134 °C, corresponding to the melting of the PP phase, respectively. It should be noted that the melting endotherms of mPOE can overlap with the glass transitions of mPOE and mPP, which are observed at around −35 °C and 0 °C in tan δ peaks, as will be discussed in the next section. Therefore, the integration of this peak for obtaining the melting enthalpy of mPOE crystals was performed at a temperature range of 0~65 °C. Additionally, two melting peaks observed at high-temperature regions indicate that the mPP has two different types of crystals and that the crystal structures are affected by the crosslinking. However, deeper studies are needed to understand this behavior clearly. Figure 3b shows two exothermic peaks at around 24.1~39.1 °C and 94.9~100.7 °C due to the crystallization of the PP phase and the POE phase, respectively. It is obvious from the DSC results that the blend has a phase-separated structure, and the *T*_m_, Δ*H*_m_, and *T*_c_ of each component polymers are lowered in the crosslinked blend as compared to those values in the neat blend. The lowering in those values indicate that crystallizations of the semicrystalline mPP and mPOE are restricted because the crosslinking caused a decrease in the molecular mobility of polymer chains [25,36]. Such changes in the thermal transition of the component polymers in the blends indicate that both mPP and mPOE participated in the reaction with PETMP and the formation of a crosslinked network.

It is also to be noted that the melting enthalpy of the PP phase, which is associated with the degree of crystallinity, is larger than that of the POE phase in the blends, and that the exotherm peaks for the PP phase are much sharper than those for the POE phase. These indicate that the PP phase has a higher degree of crystallinity and a faster melt crystallization rate than the POE phase, which affect the shape fixing of this blend sample as will be discussed in the following section.

### 3.3. Dynamic Mechanical Properties

Figure 4a,b show the variation of the storage modulus (*E*′) and tan δ with temperature for the neat blend and the blend with a different amount of PETMP, respectively. It is evident from Figure 4a that the blend with PETMP shows a higher *E*′ as compared to the neat blend over the whole temperature range examined here. The drop in *E*′ at around −50 °C and 0 °C is ascribed to the glass transition of mPOE and mPP, respectively, as will be confirmed by the tan δ curves shown in Figure 4b. It is also to be noted that *E*′ for the neat blend decreases continuously, to the point where the material flows like a viscous liquid at high-temperature regions (higher than *T*_m_), while the blend with PETMP exhibits a persistent rubbery plateau, confirming that the material formed a crosslinked structure [25,30]. The *E*’ value at the plateau region (at 150 °C), which is correlated with the degree of crosslinking of the samples shown in Table 2, increases from 0.11 to 0.53 when the PETMP content increases from 0.5 to 1.0 phr, and the value is leveled off with a further increase in the PETMP content. It is also to be noted that the crosslinked blends show steep drops in *E*′ at around 50 °C and 120 °C, which correspond to *T*_m_ of the POE and the PP phase in the blend, respectively. The modulus drop at 120 °C is more significant as compared to that observed at 50 °C because of the higher degree of crystallinity of PP than that of POE, as observed in the DSC analysis.

Figure 4b shows the variation in tan δ with temperature. It is thought that the peaks appearing at around −35 °C and 0 °C are due to the glass transition of the mPOE and mPP phases, respectively. The *T*_g_ values of each polymer in the samples are shown in Table 2. It is obvious that the *T*_g_ values of each component polymer are slightly higher in crosslinked blends as compared to those in neat blends, which indicate that the mobility of the polymer chains is restricted due to crosslinking.

### 3.4. Tensile Properties

Figure 5 presents the stress–strain curves of the neat blend and the blend crosslinked with different contents of PETMP. The tensile strength (σ_b_), the modulus at a 100% and 200% strain (*E*’_100_ and *E*’_200_), and the elongation-at-break (ε_b_) values obtained from the curves are shown in Table 3. It is obvious that the crosslinked blend has higher σ_b_, E_100_, and E_200_ as well as ε_b_ as compared to the neat blend. The neat mPP/mPOE blend has a σ_b_ of 4.8 MPa, an E_100_ of 4.3 MPa, and an ε_b_ of 237%, whereas those values increase to 14.9 MPa, 6.0 MPa, and 530% for the blend crosslinked with 1.0 phr PETMP, respectively. Such enhanced tensile properties in the mPP/mPOE blend with PETMP are ascribed to their crosslinked structure as well as intermolecular covalent bonding between the component polymers in the immiscible PP/PE blend mediated by the thiol–anhydride reaction of each component polymer with PETMP, as discussed in the above section.

### 3.5. Reprocessability

The reprocessability and recyclability of the crosslinked polyolefin blend were evaluated with a sample crosslinked with 1.0 phr of PETMP. The sample was cut into small pieces and then compression-molded at 180 °C for 15 min, and this process was repeated three times. As illustrated in Figure 6a, the recycled samples could be molded into a film with uniform thickness. The tensile properties of the recycled samples were maintained at almost the same level, with a slight increase in the stiffness during the repeated processing as shown in Figure 6b. The reprocessability results indicate that thiol–thioester exchange reactions occur throughout the network upon heating and the topology of the network is rearranged [30], which are described schematically in Figure 7.

Additionally, an increase in the stiffness was observed in the reprocessed samples as compared to the original sample. This can be attributed to the thermal decomposition of the polymers during thermal processing for prolonged times. At present, however, we could not obtain sufficient information to explain this behavior. More detailed studies will be performed in the future.

### 3.6. Shape-Memory Effect

The above results show that the crosslinked semicrystalline mPP/mPOE blends have two crystalline melting transitions at distinctly different temperatures and that there are abrupt drops in the modulus at *T*_m_ of each phase, which enables the crosslinked polyolefin blend to exhibit triple-shape-memory effects. The triple-shape-memory performance was evaluated by a cyclic thermomechanical test under control force mode as described in the experimental section, and the deformation recovery results for the blend crosslinked with 1.0 phr PETMP during three consecutive cycles are demonstrated in Figure 8.

The shape fixing (*R*_f_) and shape recovery ratio (*R*_r_) were determined using the average values from the second and third cycles because results obtained from first cycles are commonly affected by the thermomechanical history of the sample. During the second cycle, for example, the sample was elongated up to a strain of 60% under a tensile stress of 0.03 MPa at 140 °C, and then, upon cooling the sample to 80 °C, the shape was stabilized at a strain of 40.2%, which was used to determine the shape fixity ratio (*R*_f_) during shape changes from A to B. Then, the strain was increased to 150% when a stress of 0.7 MPa was applied to shape B at 80 °C, and then it was stabilized at 135% upon cooling the sample to 0 °C, from which the *R*_f_ value during the shape change from B to C was obtained. The strain decreased to 67.8% upon heating the sample to 80 °C (shape change from C to B), and then it was further decreased to 12.5% upon heating to 140 °C (shape change from B to A), from which the *R*_r_ values during each heating step were determined, respectively. The *R*_f_ and *R*_r_ values determined using Equations (1)–(4) described in the experimental section are shown in Table 4. As shown in the table, R_f_(C) is higher than R_f_(B), which is due to the faster crystallization rate of PP than POE in the blend. Shape recovery occurred step-wise by raising the temperature above *T*_m_ of the POE and PP phases in the blend consecutively, and *R*_r_ (B→A), where the shape changed from B to A, is higher than *R*_r_ (C→B), where the shape changed from C to B. This is attributed to higher entropy gains of the polymer chain during heating at higher temperature. With the increasing PETMP content from 0.5 to 1.0 phr, i.e., with increasing crosslink density, the R_r_ values increased, but were leveled off by further increasing the PETMP content.

Figure 9a demonstrates the triple-shape behavior of the mPP/mPOE blend crosslinked with 1.0 phr of PETMP. The two different temporary shapes (shape B and shape C) could be obtained by proper temperature programming, i.e., the first temporary shape (shape B) was obtained by deforming the original rectangular shape (shape A) to shape B at 150 °C and a subsequent cooling to 80 °C in stress-free state, and the other temporary shape (shape C) was obtained by deforming shape B to shape C at 80 °C and subsequently cooling it to 0 °C. Shape recovery was observed to occur in reverse order, i.e., shape C recovered to shape B upon heating at 80 °C, and then shape B recovered to its original rectangular shape upon further heating to 150 °C.

Figure 9b shows that the original shape of the sample can be reconfigured into another permanent shape at high temperature. As shown, a rectangular shape, which was initially obtained by compression molding, could be reconfigured into a permanent S shape when the original sample was deformed at 180 °C for 30 min. Afterward, a triple-shape-memory effect was observed at the same temperature programming as employed in Figure 9b.

## 4. Conclusions

This work demonstrates that a thermally reprocessable chemically crosslinked polymer with shape reconfigurability and a triple-shape-memory effect can be prepared by the simple reactive melt blending of two different semicrystalline maleated polyolefins (mPP and mPOE) with a small amount of PETMP and a transesterification catalyst. The crosslinked blend showed superior mechanical properties as compared to a simple blend, and the mechanical properties were maintained after repeated thermal molding due to a thermal-activated exchange of dynamic thioester bonds existing in the network. This study suggests that mixtures of thermoplastic semicrystalline polyolefins generated abundantly in daily life can be transformed into a high-performance and value-added material through a simple and economical process.

## Figures and Tables

**Figure 1 polymers-16-02714-f001:**
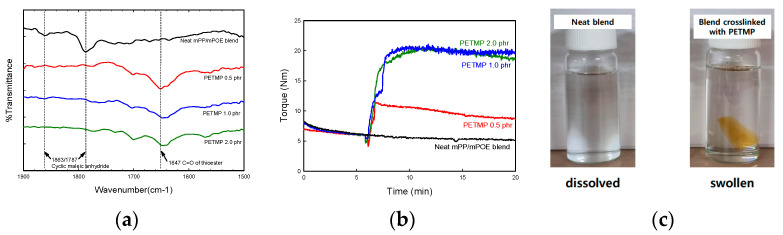
(**a**) FT-IR spectra; (**b**) torque-time curve; (**c**) solubility test.

**Figure 2 polymers-16-02714-f002:**
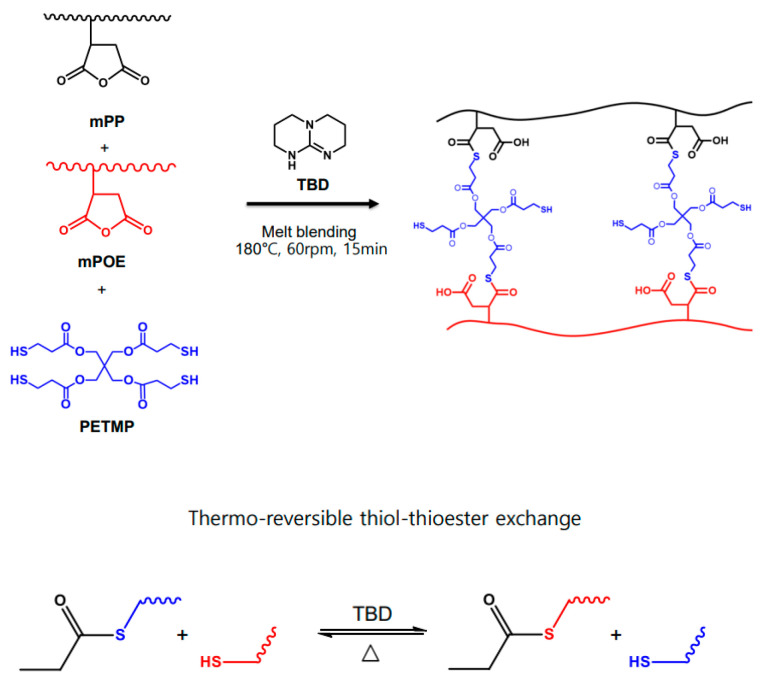
Chemical structure of the crosslinked network obtained from the crosslinking of mPP/mPOE blend with PETMP in the presence of a TBD catalyst and possible thiol–thioester exchange reactions occurring in the network under thermal activation.

**Figure 3 polymers-16-02714-f003:**
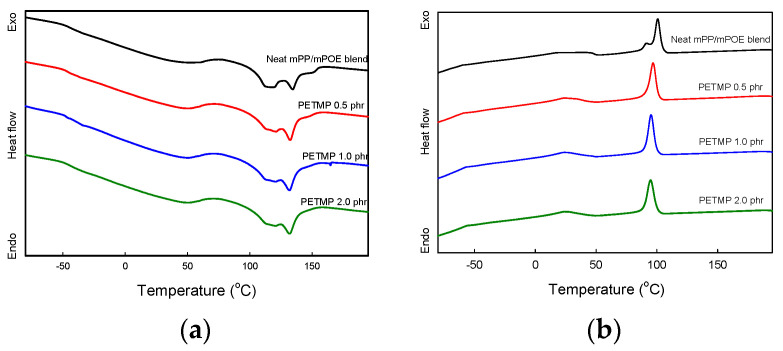
DSC thermograms of samples. (**a**) Heating scan; (**b**) cooling scan.

**Figure 4 polymers-16-02714-f004:**
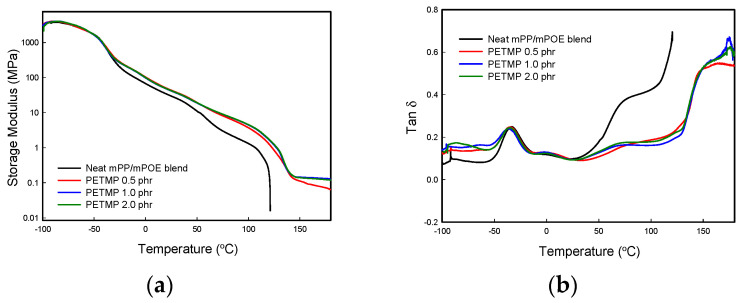
Variation in (**a**) storage modulus (*E*’) and (**b**) tan δ with temperature for the neat mPP/mPOE blend and the blend with PETMP.

**Figure 5 polymers-16-02714-f005:**
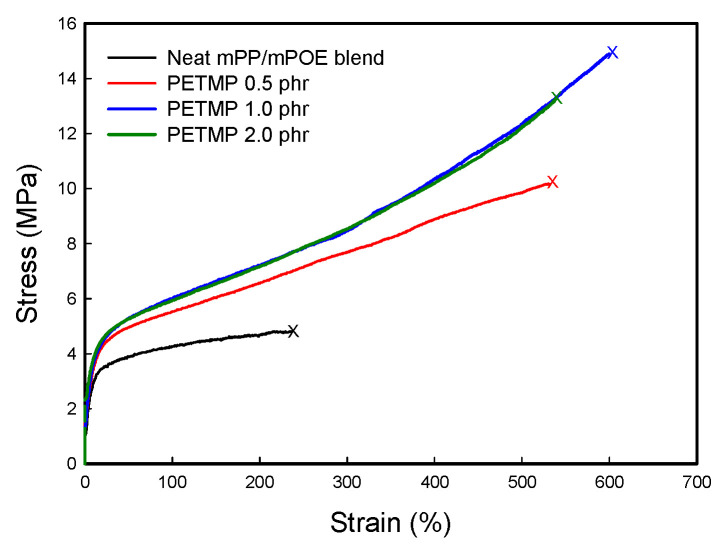
Stress–strain curves of samples.

**Figure 6 polymers-16-02714-f006:**
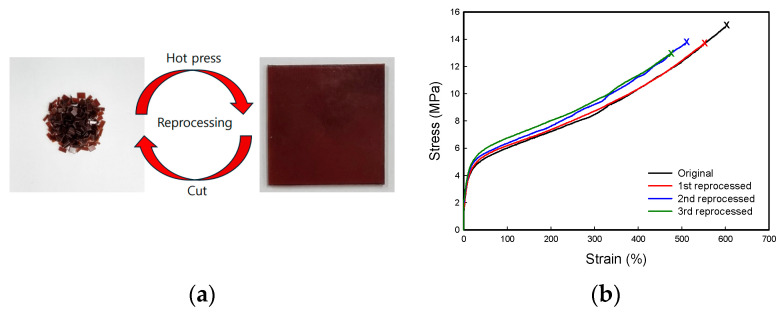
(**a**) Photographs showing the reprocessability of a blend sample crosslinked with 1.0 phr PETMP; (**b**) stress–strain curves of the same sample after being recycled 3 times.

**Figure 7 polymers-16-02714-f007:**
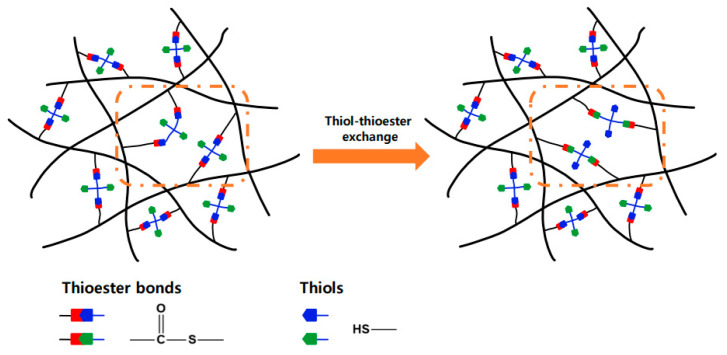
Topological rearrangement of the network via thermal-activated thiol-thioester exchange reactions.

**Figure 8 polymers-16-02714-f008:**
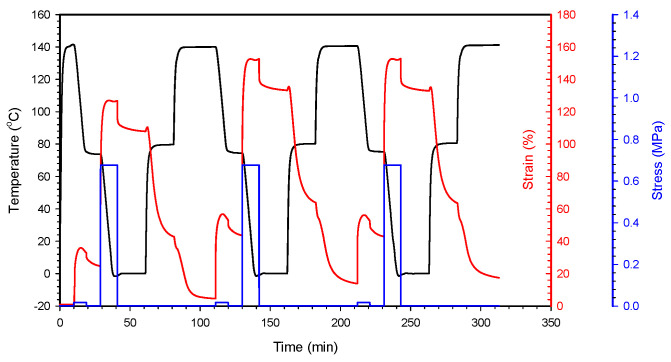
Deformation recovery during three consecutive cyclic thermomechanical tests for mPP/mPOE blend crosslinked with 1.0 phr PETMP.

**Figure 9 polymers-16-02714-f009:**
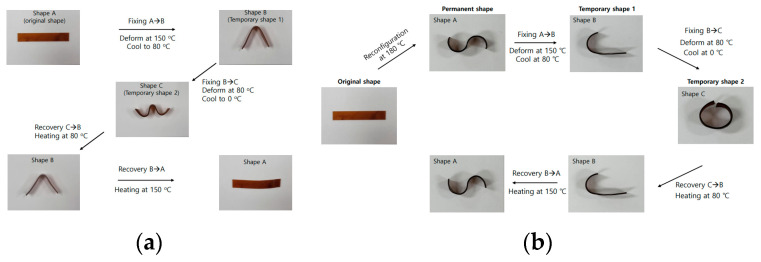
(**a**) Triple-shape-memory behavior; (**b**) shape reconfigurability of mPP/mPOE blend crosslinked with 1.0 phr PETMP.

**Table 1 polymers-16-02714-t001:** Thermal characteristics of samples.

Sample	POE Phase			PP Phase		
	*T*_m_ (°C)	Δ*H*_m_ (J/g)	*T*_C_ (°C)	*T*_m_ (°C)	Δ*H*_m_ (J/g)	*T*_C_ (°C)
Neat mPP/mPOE blend	52.8	5.25	39.1	115.7, 134.7	19.9	100.7
PETMP 0.5 phr	50.5	4.80	24.2	121.0, 132.5	19.8	96.9
PETMP 1.0 phr	50.3	4.22	24.1	120.7, 132.0	19.1	95.3
PETMP 2.0 phr	50.4	4.06	24.1	120.8, 132.1	18.3	94.9

**Table 2 polymers-16-02714-t002:** Storage modulus (*E*’) at rubbery plateau region (*T* = 150 °C) and *T*_g_ obtained from tan δ peak maximum of samples.

Sample	*E*’ at 150 °C (MPa)	*T*_g_ of mPOE (°C)	*T*_g_ of mPP (°C)
Neat mPP/mPOE blend	-	−35.5	1.01
PETMP 0.5 phr	0.11	−35.1	1.09
PETMP 1.0 phr	0.53	−34.6	1.14
PETMP 2.0 phr	0.54	−34.4	1.15

**Table 3 polymers-16-02714-t003:** Tensile properties of samples.

Sample	*E*_100_ (MPa)	*E*_100_ (MPa)	σ_b_ (MPa)	ε_b_ (%)
Neat mPP/mPOE blend	4.3 ± 0.1	4.7 ± 0.1	4.8 ± 0.2	237 ± 70
PETMP 0.5 phr	5.5 ± 0.1	6.6 ± 0.2	10.2 ± 0.3	531 ± 70
PETMP 1.0 phr	6.0 ± 0.1	7.2 ± 0.2	14.9 ± 0.3	600 ± 50
PETMP 2.0 phr	6.1 ± 0.1	7.2 ± 0.2	13.2 ± 0.3	539 ± 50

**Table 4 polymers-16-02714-t004:** Shape fixing (*R*_f_) and shape recovery ratio (*R*_r_) of mPP/mPOE blend crosslinked with different contents of PETMP.

Sample	*R*_f_ (B) (%)	*R*_f_ (C) (%)	*R*_r_ (C→B) (%)	*R*_r_ (B→A) (%)
PETMP 0.5 phr	67.3	89.8	64.2	69.5
PETMP 1.0 phr	67.0	90.1	70.8	78.7
PETMP 2.0 phr	67.2	90.1	71.5	79.5

## Data Availability

Data are contained within the article.

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
