# Peer review of "Covalent Adaptable Network of Semicrystalline Polyolefin Blend with Triple-Shape Memory Effect"

_polymers, 2024, doi:10.3390/polym16192714_

Round 1

Reviewer 1 Report

Comments and Suggestions for Authors

This work demonstrated that a thermally reprocessable chemically crosslinked polymer having triple-shape memory effect can be prepared. 

A good article that can be published after providing additional data:

1. Since the practical result of the work consists in the course of a complex of reversible chemical reactions, it is necessary to clearly describe their entire closed cycle. Without a chemical component, the material of the article does not give a complete information of the idea of creating the resulting polymer products.

2. A remark directly related to the first one. The authors very correctly used the method of dynamic mechanical analysis to describe the properties of the material, however, in my opinion, it is necessary to present on the graph not only the evolution of the storage modulus (E′), but also the evolution of the values of the tangent of the angle of mechanical losses. Combining these two parameters on one graph will help to understand and describe the dynamics of molecular movements in the created polymer material much more deeply.

Author Response

Comment #1. Since the practical result of the work consists in the course of a complex of reversible chemical reactions, it is necessary to clearly describe their entire closed cycle. Without a chemical component, the material of the article does not give a complete information of the idea of creating the resulting polymer products.

Response to comment #1 :

Thank you for the comments.  In order to provide information for our idea of creating the polymer products, chemical structure of the prepared network and topological rearrangement in this network  accompanied with dynamic exchange reaction were presented in Figure 2 (page 4) and Figure 7 (page 8), respectively. 

Comment #2. A remark directly related to the first one. The authors very correctly used the method of dynamic mechanical analysis to describe the properties of the material, however, in my opinion, it is necessary to present on the graph not only the evolution of the storage modulus (E′), but also the evolution of the values of the tangent of the angle of mechanical losses. Combining these two parameters on one graph will help to understand and describe the dynamics of molecular movements in the created polymer material much more deeply.

Responses to comment #2 :

Thank you for the comment.  As suggested by the reviewer, tan delta curves obtained by dynamic mechanical analysis are presented in Figure 4 (b). And, the values of storage modulus at rubbery plateau region (T > Tm) and Tg obtained from tan delta peak maximum are tabulated in Table 2. Discussions for these results are also added [Line #239-246, page 6].

Reviewer 2 Report

Comments and Suggestions for Authors

The authors report blending of two polyolefins and subsequent dynamic crosslinking into a covalent adaptable network. The authors further claim triple shape memory, leveraging the crystalline phases of either of the polyolefins and the dynamic covalent bonds.

The authors refer to triple (3) shape memory, while only two (2) temporary shapes have been fixed sequentially.

The introduction should clearly discuss how shape memory based on dynamic covalent chemistries works and what the limitations are. In contrast to physical shape memory using crystallization or vitrification, reactive shape memory does not fix the polymer chain mobility, but simply fixes the network topology.

The melting and crystallization of mPOE extends from -50 to 50 °C. This should at least be commented on in the text to help the reader interpret the DSC thermograms. The DMA results confirm that already from -50 °C the properties are decreasing drastically due to the melting of the first crystalline phase.

The enthalpy under the very broad peak doesn’t look much smaller in area than the narrow mPP melting peak. Please, verify if the areas were integrated correctly. I don’t expect an order of magnitude.

Moreover, the melting of mPP happens in two steps after processing. How does this compare to the second heating in DSC? Are the two crystal types a consequence of the processing or a result of the mPP structure/composition?

How does DMA tell you whether a material is crosslinked or not?

After reprocessing, the behaviour beyond the yield point (elastomeric behaviour) shows stiffening with additional processing cycles. Spectroscopic data and calorimetry should show the extent of the changes in chemical structure and thermophysical properties.

The sample was elongated by 60% (B) and retained 40.2% elongation upon cooling. This is a shape fixity of 67%, while 82.3% is tabulated. The sample was then elongated to 150% (C) and retained 135%, which comes down to 90% fixity (89.2% tabulated). The shape recovers then to 67.8%, so from the remaining 135% elongation 67.2% is recovered, compared to the 94.8% (135% - 40.2%) that was available. That’s 70.8% and not the tabulated 80.2%. Please, verify all calculations carefully!

Also, explain the reader how to calculate the recovery ratio for the second shape recovery (first fixed shape) from B to A. It is not explained.

Comments on the Quality of English Language

Grammar and spelling checking are required for the entire manuscript.

Often there are double spaces after punctuation, especially in the introduction.

Author Response

Comment #1 : The authors refer to triple (3) shape memory, while only two (2) temporary shapes have been fixed sequentially.

Response to comment #1

‘Triple’ in the triple shape memory refers to the two (2) temporary shape plus one (1) permanent shape.  Such triple shape memory polymers having the two temporary shape which can be recovered sequentially with temperature rise have also been reported in the literature [ref # 10-26].

Comment #2: The introduction should clearly discuss how shape memory based on dynamic covalent chemistries works and what the limitations are. In contrast to physical shape memory using crystallization or vitrification, reactive shape memory does not fix the polymer chain mobility, but simply fixes the network topology.

Response to comment #2

Features of the dynamic covalent chemistry based SMPs are added in Introduction in detail [Line #62-70, page 2], and last paragraph of the Introduction was a little modified to express the motivation and goal of the present study more clearly. [Line #71-75, page 2]  And, a result demonstrating shape reconfigurability of the polymer we prepared, which is one of the important features observed in other dynamic covalent chemistry based SMPs, is added as Figure 9(b) [Line #336-348, page 9-10]

Comment #3 : The melting and crystallization of mPOE extends from -50 to 50 °C. This should at least be commented on in the text to help the reader interpret the DSC thermograms. The DMA results confirm that already from -50 °C the properties are decreasing drastically due to the melting of the first crystalline phase.

Responses to comment #3

Thank you for the comments.  In order to interpret the DSC thermgram more clearly, tan δ curves of the sample was added (Figure 4b), which showed that glass transition of mPOE and mPP appeared at around -35 oC and 0 oC, respectively.  So, the melting endotherm of mPOE in DSC thermogram can be overlapped with the glass transitions of mPOE and mPP.   Integration of the peak was performed with temperature range of 0 ~ 65 oC. [Line #187-194, page 5] And, the decrease in modulus at around -50 oC shown in Figure 4(a) is not due to melting of POE crystal but due to glass transition. [Line #220-222, page 6]  

Comment #4 : The enthalpy under the very broad peak doesn’t look much smaller in area than the narrow mPP melting peak. Please, verify if the areas were integrated correctly. I don’t expect an order of magnitude.

Responses to comment #4

Integration value of the area of the melting peak (enthalpy) of mPOE was verified carefully and are shown in Table 1 [page 6].

Comment #5 : Moreover, the melting of mPP happens in two steps after processing. How does this compare to the second heating in DSC? Are the two crystal types a consequence of the processing or a result of the mPP structure/composition?

Response to comment #5

Thank you for the comments.  The two melting peaks of mPP indicate that the mPP has two different type of crystals, and the crystal structures of the mPP are affected by the crosslinking. More deep studies are needed to understand this behavior. [Line #192-194, page 5]

Comment #6 : How does DMA tell you whether a material is crosslinked or not?

Response to comment #6

If the polymer chains (for semicrystalline polymers) are not crosslinked, the storage modulus continues to decrease to a point where the material flows like a viscous liquid for temperatures greater than its Tm. On the other hand, crosslinked polymers show persistent rubbery plateau at this temperature region.  This behavior has been observed for other crosslinked semicrystalline polymers.[Line # 222-225, page 6]

Comment #7 : After reprocessing, the behaviour beyond the yield point (elastomeric behaviour) shows stiffening with additional processing cycles. Spectroscopic data and calorimetry should show the extent of the changes in chemical structure and thermophysical properties.

Response to comment #7

It is supposed that the increase in the stiffness for the thermally reprocessed samples is associated with thermal decomposition of the polymers during thermal molding at high temperature for prolonged time. We performed FTIR and DSC analysis for the reprocessed samples, but could not obtain any meaningful results.  More deep studies are needed to understand this behavior, which are remained as future works. [Line # 274-278, page 7]

Comment #8 : The sample was elongated by 60% (B) and retained 40.2% elongation upon cooling. This is a shape fixity of 67%, while 82.3% is tabulated. The sample was then elongated to 150% (C) and retained 135%, which comes down to 90% fixity (89.2% tabulated). The shape recovers then to 67.8%, so from the remaining 135% elongation 67.2% is recovered, compared to the 94.8% (135% - 40.2%) that was available. That’s 70.8% and not the tabulated 80.2%. Please, verify all calculations carefully!  Also, explain the reader how to calculate the recovery ratio for the second shape recovery (first fixed shape) from B to A. It is not explained.

Responses to comment #8

The shape fixity and recovery values in Table 4 were checked carefully and are corrected. [page 9]  And, explanations for the calculation of the recovery ratio for the second shape recovery were added in Experimental part [Line # 128-142, page 3].

Comment #9 :  Grammar and spelling checking are required for the entire manuscript. Often there are double spaces after punctuation, especially in the introduction.

Response to comment #9

Grammar and spellings are checked throughout the manuscript carefully.  The spaces between the sentences are corrected to be a single space.

Round 2

Reviewer 1 Report

Comments and Suggestions for Authors

Thanks for additional information. Tha manuscript can be accepted in present form

Reviewer 2 Report

Comments and Suggestions for Authors

The authors have taken into account all proposed comments. The manuscript can be considered for publication without further revisions.